# Cloud- and ice-albedo feedbacks drive greater Greenland ice sheet sensitivity to warming in CMIP6 than in CMIP5

Idunn Aamnes Mostue[1], Stefan Hofer[1,2], Trude Storelvmo[1], and Xavier Fettweis[3]

[1]Department of Geosciences, University of Oslo, Oslo, Norway
[2]School of Geographical Sciences, University of Bristol, Bristol, United Kingdom
[3]Department of Geography, SPHERES research unit, University of Liège, Belgium

**Correspondence:** Idunn Aamnes Mostue (idunnam@uio.no)

**Abstract.** The Greenland Ice Sheet (GrIS) has been losing mass since the 1990s as a direct consequence of rising temperatures and has been projected to continue to lose mass at an accelerating pace throughout the 21st century, making it one of the largest contributors to future sea-level rise. The latest Climate Model Intercomparison Project 6th phase (CMIP6) models produce a greater Arctic amplification signal and therefore also a notably larger mass loss from the GrIS when compared to the older CMIP5 projections, despite similar forcing levels from greenhouse gas emissions. However, it is also argued that the strength of regional factors such as melt-albedo feedbacks and cloud-related feedbacks will partly impact future melt and sea-level rise contribution, but little is yet known about the role of these regional factors in differences in GrIS surface melt projections between CMIP6 to CMIP5. In this study, we use high-resolution (15 km) regional climate model simulations over the GrIS performed using the Modéle Atmosphérique Régional (MAR) to physically downscale six CMIP5 RCP8.5 and five CMIP6 SSP5-8.5 extreme high-emission scenario simulations. Here, we show a greater annual mass loss from the GrIS at the end of the 21st century, but also for a given temperature increase over the GrIS, when comparing CMIP6 to CMIP5. We find a greater sensitivity of Greenland surface mass loss in CMIP6 centred around summer and autumn, yet the difference in mass loss is largest during autumn with a reduction of $27.7 \pm 9.5$ Gt/season for a regional warming of $+6.7°C$, 24.6 Gt/season more mass loss than in CMIP5 RCP8.5 simulations for the same warming. Assessment of the surface energy budget and cloud-related feedbacks suggests a reduction in high clouds during summer and autumn – in addition to enhanced cloud optical depth during autumn – to be the main drivers of the additional energy reaching the surface, subsequently leading to enhanced surface melt and mass loss in CMIP6 compared to CMIP5. Our analysis highlights that Greenland is losing more mass in CMIP6 due to two factors; 1) a (known) greater sensitivity to greenhouse gas emissions and therefore warmer temperatures, 2) previously unnotified cloud-related surface energy budget changes that enhance the GrIS sensitivity to warming.

# 1 Introduction

The Greenland Ice Sheet (GrIS) has been losing mass at an accelerating pace since the mid 1990s and is expected to continue to lose mass during the 21$^{st}$ century (Hanna et al., 2008; Fettweis et al.,2013; Mouginot et al., 2019; Nöel et al., 2021; Doblas-Reys et al.,2021). Fluctuations in the mass balance occur with variations in glacial discharge, meltwater runoff, and accumulation of snow on the ice sheet surface. However, recent mass loss is dominated by a reduction in the surface mass balance (SMB) along the edges of the ice sheet – the ablation zone – from surface melt (ME) and subsequent runoff (RU) from meltwater produced at the surface (van den Broeke et al., 2016; IMBIE2, 2020).

Due to the dark exposed bare ice during summer melt season, melt in the ablation zone is mainly driven by absorbed solar radiation (Box et al., 2012; van den Broeke et al., 2008; van den Broeke et al., 2017; Nöel et al., 2019). The displacement of mass from the ice sheet into the ocean contributes to global sea-level rise, threatening coastal ecosystems, human habitats and livelihood (Hauer et al., 2020; Doblas-Reys et al.,2021).

Clouds are of first order importance in altering the GrIS energy budget, both in the longwave (LW) radiative energy spectrum by absorbing and emitting LW, and in the shortwave (SW) radiative energy spectrum by reducing the amount of the incoming SW and thus cooling the surface. There is thus a competing influence on the SW and LW radiative energy spectra from clouds over the GrIS, which can either warm or cool the surface (Shupe et al., 2004; Bennartz et al., 2013; van Tricht el al., 2016; Hofer et al., 2019).

Arctic amplification and circulation changes have been pointed out as the main drivers of the recent SMB loss from the GrIS (Tedesco et al., 2011; Bennartz et al., 2013; Fausto et al., 2016; Tedesco et al., 2016). While Arctic amplification leads to anomalous increase in the near-surface temperatures (Screen and Simmonds, 2010), more frequent anticyclonic circulation conditions lead to a reduction in clouds (Hofer et al., 2017), and enhanced melt-albedo feedbakcs (Tedesco et al., 2011; Box et al.,2012).

The latest Coupled Model Intercomparison Project (CMIP) 6$^{th}$ phase models produce a greater Arctic Amplification signal and therefore also a notably larger mass loss from the GrIS when compared to older CMIP5 projections, despite using nominally comparable forcing scenarios (O'Neil et al., 2016). CMIP6 models have also shown to project more Arctic precipitation compared to CMIP5 at the end of the 21$^{st}$ century (McCrystall et al., 2021), which can impact the surface albedo and further the surface mass balance (Box et al., 2022). However, a study by Hofer et al. (2020) showed that the rainfall projections between CMIP5 and CMIP6 regional climate model output over Greenland only starts diverging from the year 2070 onward, whereas the surface mass balance start to diverge from year 2020 onward. In addition, the two model groups also have different regional feedbacks, such as melt-albedo feedback and cloud-related feedbacks which will partly impact future melt and sea-level rise contribution by altering the energy available for melt at the surface. Both the difference in Arctic warming levels and regional feedbacks will influence future mass loss. Yet, little is known about the relative importance of these regional factors when comparing CMIP6 to CMIP5 over Greenland.

The work of this study builds on Mostue, I. Aa. (2022). We analyse regional climate model outputs with the main focus on comparing future projections at a given warming level. In this way we can disentangle whether the difference in melt and mass

loss come from a greater sensitivity at a given temperature or just from the fact that CMIP6 models warm more in absolute terms over Greenland and the Arctic.

## 2 Methods

### 2.1 Driving mechanisms of surface melt

Significant amounts of extra energy are needed for melt-induced surface mass loss over the GrIS (van den Broeke et al., 2008;
Franco et al., 2013). Therefore the analysis of this study focuses on the long term changes in radiative Surface Energy Budget (SEB) components over the ice sheet. With positive orientation downwards to the surface, the balance can be described through Equation (1), where a surplus in the SEB will give rise to surface melt (ME). The net longwave radiation ($LW_{net}$) constitute of the longwave radiation emitted downwards to the surface (LWD), and from the surface (LWU). The net shortwave radiation ($SW_{net}$) depends on the incoming solar radiation on top-of-the-atmosphere (i.e. shortwave down – SWD), the surface albedo
($\alpha$), as well as the influence of clouds, aerosols etc. on the transmissivity of the atmosphere (i.e. altering the amount of incoming SWD reaching the surface).

$$
\begin{aligned}
ME &= LWD - LWU + SWD(1 - \alpha) + SHF + LHF + GHF \quad [\text{Wm}^-2] \\
&= LW_{net} + SW_{net} + SHF + LHF + GHF \quad [\text{Wm}^-2]
\end{aligned}
\tag{1}
$$

The surface mass balance (SMB) defines the difference in accumulation (i.e. through precipitation – P), and ablation (i.e. through sublimation – SU, erosion of snow by wind – E, and meltwater ruoff – RU) at the ice sheet:

$$SMB = PR - SU - E - RU \quad [\text{Gt yr}^-] \tag{2}$$

In the interior of the GrIS – the accumulation zone – the snow pack is highly reflective to any incoming solar radiation (i.e it has a high albedo), and so any variation in the absorbed energy budget over the accumulation zone is primarily controlled by the longwave radiation fluxes (LW) (Charalampidis et al., 2015 ;van den Broeke et al., 2017).

In contrast to the high albedo in the accumulation zone, the ablation zone around the edges of the ice sheet experience bare
ice exposure during the summer melt, thus absorption of SWD is enhanced. Hence, in this area the surface energy budget is primarily controlled by shortwave (SW) radiation fluxes when bare ice is exposed during summer melt season (van de Wal et al., 2005; van den Broeke et al., 2011).

### 2.2 Data

For the analysis of this study we use data based on the same simulations as were used for segments of the work done by Hofer et
al. (2020). They used a set of 11 high-resolution (15 km) regional climate model simulations over the GrIS. These simulations were conducted by using the regional climate model Modéle Atmosphérique Régional (MAR) to physically downscale six

CMIP5 and five CMIP6 models (Table 1), using the RCP 8.5 and SSP5-8.5 high-emission scenarios, respectively. Hofer et al. (2020) base their CMIP5 model selection on the Ice Sheet Model Intercomparison Project for CMIP6 (ISMIP6), fully described in Barthel et al., 2020. A 'top three' CMIP5 model ensemble was selected following the ISMIP6 protocol (i.e. i. the model must provide 6 h outputs to provide as input to MAR at its lateral boundaries, ii. the model must provide 6 h outputs for RCP2.6 and RCP8.5 scenario projections, iii. with regards to historical atmospheric, surface and subsurface ocean metrics, the model must lie in the upper half of 33-model ensemble, iv. All climate change metrics of the model must lie within the two interquartile range of the multi-model median of the normalised projected change over the Antarctica and Greenland). An additional three models were picked to to maximise the projected diversity of the end to century climate change projections i.e. to capture the full diversity of the ensemble. Hofer et al. (2020)'s selection of the top CMIP6 models was limited by model availability at the time being, with only five of the seventeen available models meeting the first requirement of 6 h output. However, all five models were included in the CMIP6 model ensemble, as Hofer et al. (2020) found them to acceptably represent the model mean, minimum, and maximum of the full ensemble.

For this study we use the same 11 MAR simulations running from 1950 to 2100. However, in contrast to previous studies we calculate different anomalies for various Greenland climate variables, i.e. near-surface temperature (2-m temperature), SMB, cloud cover and SEB components based on the 1961–1990 mean state of each model simulation.The GrIS is assumed to have been in stable state during the thirty-year period from 1961–1990 (van den Broeke et al., 2016), thus why this was chosen as our reference period. Furthermore, we focus on the physical drivers of enhanced GrIS mass loss in CMIP6 at a given level of warming, which have not been studied in detail before.

We use the anomalies for projecting any yearly or seasonal spatially averaged change over the GrIS surface for a given regional temperature increase, and compare these projections produced by CMIP5 and CMIP6. These anomalies were calculated individually for each of the MAR simulations forced by the eleven GCMs, before being averaged over all six MAR simulations of CMIP5 models (hereafter MAR CMIP5) and over all five MAR simulations of CMIP6 models (hereafter MAR CMIP6). As the focus of our analysis is on the response and changes over the ice sheet area of Greenland only, we mask out every modelled pixel containing less than 10 % ice cover. With a 10 % ice cover mask that does not change over time, we expect our SMB reduction to be slightly overestimated compared to a dynamic ice mask, but recent research indicated this error to be somewhere between 1 % and 6 % (Kjeldsen et al., 2020; Hansen et al., 2022).

Additionally, we look at a twenty-year averaged period of $\sim$4°C (2-m temperature) regional warming over Greenland to seek any potential differences in how changes in cloud cover, radiation and surface mass flux variables are spatially distributed over the ice sheet, so to gain further insight into the spatial patterns of changes caused by rapid Greenland warming. The individual CMIP models warm at different rates, thus do not reach the same temperature by the end of the century. We therefore look at a threshold temperature to be able to compare all models for the same temperature increase. We decided on a 4°C warming as this is the highest temperature rise for a twenty-year averaged period reached by all CMIP5 and CMIP6 models. These twenty-year warming periods were calculated for a seasonal mean, and for each of the eleven MAR simulations individually by creating a moving average, with a centred window of twenty-years over the near-surface warming time series. This allowed us to compute the arithmetic mean along the time series, where a twenty-years ($\pm$ 10 years around each step) averaged for each year in the

**Table 1.** Forcing fields used to perform MAR simulations, historical periods and future scenarios of the simulations, and CMIP phase.

| Forcing fields | Historical simulation | Future Scenario | CMIP phase |
|---|---|---|---|
| ACCESS1.3 | historical (1850–2005) | RCP8.5 | 5 |
| CSIRO-Mk3-6-0 | historical (1850–2005) | RCP8.5 | 5 |
| HadGEM2-ES | historical (1850–2005) | RCP8.5 | 5 |
| IPSL-CM5A-MR | historical (1850–2005) | RCP8.5 | 5 |
| MIROC5 | historical (1850–2005) | RCP8.5 | 5 |
| NorESM1-M | historical (1850–2005) | RCP8.5 | 5 |
| CESM2 | historical (1850–2014) | SSP5-8.5 | 6 |
| CNRM-CM6-1 | historical (1850–2014) | SSP5-8.5 | 6 |
| CNRM-ESM2-1 | historical (1850–2014) | SSP5-8.5 | 6 |
| MRI-ESM2-0 | historical (1850–2014) | SSP5-8.5 | 6 |
| UKESM1-0-LL | historical (1850–2014) | SSP5-8.5 | 6 |

time series is returned. Then, by picking the year that returns a temperature closest to our designated near-surface temperature we found the twenty-years time interval for each model of similar averaged warming (see Table T2 in supplementary for an overview of the individual warming periods of each MAR simulation). After picking a twenty-year warming period for each model we averaged over all six MAR simulations of CMIP5 models and over all five MAR simulations of CMIP6 models separately.

### 2.3 Modèle Atmosphérique Régional

As most of the increase in melt occurs in the narrow ablation zone or through an expansion of the ablation zone, models used to project future changes in SMB must be able to represent the dynamics at the local spatial scale of this area. The Modèle Atmosphérique Régional (MAR) is a Polar Regional Climate model. The grid extent, projection and resolution of MAR has been adapted specifically to the GrIS, which makes it capable of capturing regional changes of the SMB over the ice sheet (Fettweis et al., 2011, 2017). For this study, we use the MARv3.9.6, previously evaluated by Delhasse et al. 2020. MAR is a hydrostatic primitive equation model, consisting of a three-dimensional atmospheric model, coupled to a one-dimensional energy-balance based surface- and vegetation- model SISVAT (Soil Ice Vegetation Atmospheric Transfer) (Gallee and Schayes, 1994; De Ridder , 1998; Fettweis et al., 2013). SISVAT models the exchange between the atmosphere and surface. It is multilayered and subdivided into a soil-vegetation module, and a snow-ice module, where the latter is based on the snow model CROCUS (Brun et al.,1992; Vionnet et al., 2012).

The MAR Greenland setup covers an integration domain stretching from -88,4° W,5.1°E longitudinal extent and 54.89°S, 85.92°N latitudinal extent (see Figure S1 in supplementary). A 15x15 km spatial resolution was used, covering a domain of 115x210 [longitude, latitude] gridpoints. The MAR Greenland setup was ran with a vertical resolution of 24 pressure-

ratio levels, with the model top pressure set at 0.1hPa. A six-hourly input on specific humidity, u- and v-wind components, temperature and sea-level pressure, as well as daily input on sea-surface temperature and concentration was provided at the lateral boundaries of MAR.

For the simulations used in this study, the MAR was ran in 'community mode' meaning that a member is started every 5 years over 1950–2090 and initialised by the snowpack simulated for this year by former MARv3.9 based on simulations using the same GCM as forcing and where each member simulates at least 15 years. As the period simulated by each member covers at least two members initialised at different years (5 and 10 years ago), the retained years of each member was chosen to be independent of the initial conditions i.e. to have difference of SMB, runoff and refreezing lower than 1 GT/yr between the different members for the same year. Due to the high liquid water content allowed in MAR, a snowpack can lose quickly (∼10 years) its capacity to retain meltwater as it becomes too dense. Further, these simulations resolve the 30 first meter of snow. A layer is automatically added/removed at the bottom if the total snow is < 29 m or > 31 m. The maximum liquid water content in MAR is 7% (Lefebre et al., 2003).

Finally, the MAR model physics and resolution remained unchanged across the downscaling for all eleven GCMs.

## 3 Results

### 3.1 GrIS Surface Mass Balance change

Annual, summer (JJA) and autumn (SON) seasonal SMB, melt and runoff anomalies [Gt/season] are presented in Figure 1 as functions of near-surface temperature anomalies [°C] over the GrIS. In year 2100 we observe that MAR CMIP6 temperatures extend beyond those from MAR CMIP5 on the x-axis, yielding a + 1.4°C greater warming over the GrIS than in MAR CMIP5 annually (+1.8°C in summer and autumn). Consequently, MAR CMIP6 reaches a higher melt- (pink) and runoff- (purple) anomaly and subsequently a greater surface mass loss (green), partly due to a higher near-surface temperature anomaly than what is found for MAR CMIP5. However, we also find a greater sensitivity of the GrIS SMB reduction in MAR CMIP6 even when looking at the same warming. In Figure 1 a) we see that MAR CMIP6 projects a higher annual melt and runoff anomaly for a given warming, and subsequently a greater surface mass loss (green, SMB), when compared to MAR CMIP5. This clearly suggests that parts of the greater mass loss in MAR CMIP6 over the GrIS are driven by a difference in SMB sensitivity for a given temperature change between the ensembles.

Due to the geographical position of Greenland, the timing of the melt season is closely connected to the seasonal changes in solar radiation (Fettweis et al., 2011), and we expect most of the surface melt to occur over the three summer months June, July, August (JJA). During summer (JJA), we observe from Figure 1 b) the largest absolute contribution of melt, runoff and SMB reduction for both MAR CMIP5 and MAR CMIP6 in a warming climate. Conversely, from Figure 1 c) we find that the largest difference in projected melt, runoff and SMB between MAR CMIP5 and MAR CMIP6 comes from autumn (SON). Here, MAR CMIP6 projects a surface mass loss reduction of 27.7 ± 9.5 Gt/season for a temperature increase of + 6.7 °C (see Table T1 in supplementary), 24.6 Gt/season more mass loss than for MAR CMIP5 at the same temperature increase (see Table T1 in supplementary). For winter (DJF, Figure S2 in supplementary) and spring (MAM, Figure S3 in supplementary) the

difference in SMB between MAR CMIP5 and MAR CMIP6 is negligible and will not be further discussed. This suggests that
the main driver of the greater mass loss sensitivity in MAR CMIP6 compared to MAR CMIP5 in the annual mean stems from
the difference in sensitivity in autumn.

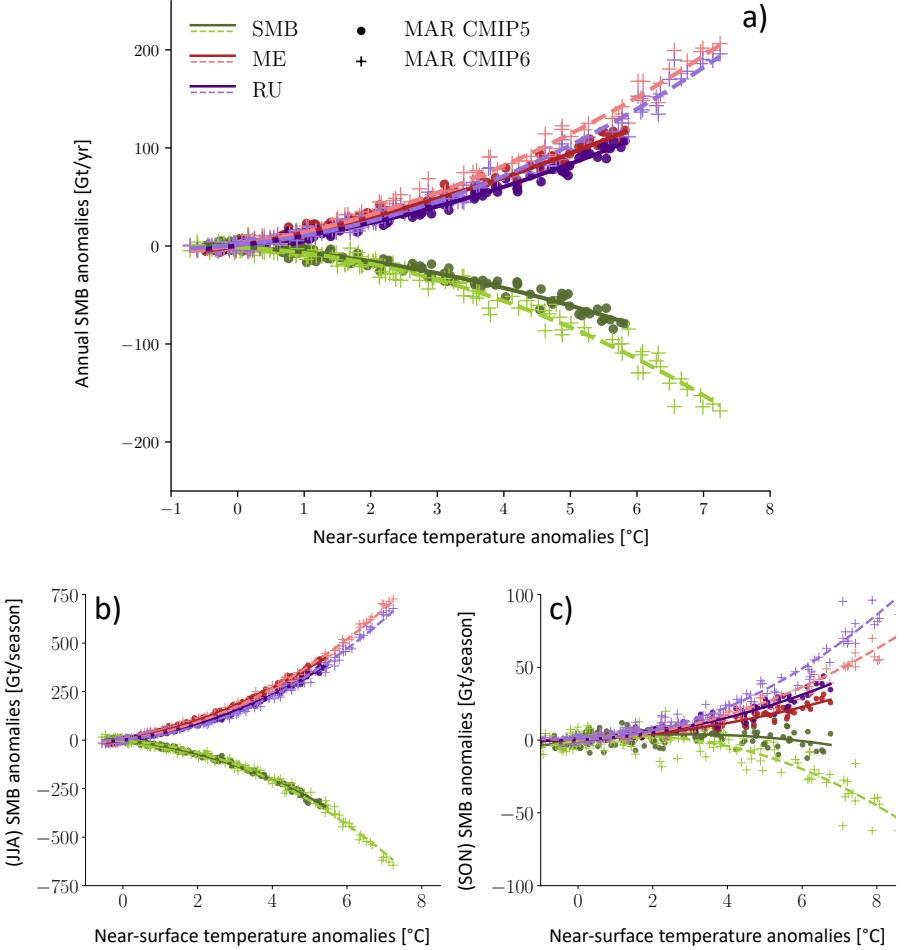

**Figure 1. SMB, melt and runoff anomalies [Gt/yr] over the GrIS as a function of Greenland near-surface temperature anomalies [°C]. a)** Annual SMB, melt- (ME), and runoff anomalies (RU) [Gt/yr] over the GrIS as a function of annual near-surface temperature anomalies [°C] over Greenland from MAR CMIP5 (dots) and MAR CMIP6 (crosses), with regression drawn in solid lines for MAR CMIP5 and scattered lines for MAR CMIP6. All anomalies are related to the thirty-years averaged reference period (1961–1990). **b)** and **c)** same as **a)**, but for a seasonal mean of summer (JJA) and autumn (SON), respectively.

## 3.2 Cloud contribution to Surface Energy Budget change

To assess where the difference in SMB for a given warming between MAR CMIP5 and MAR CMIP6 comes from, we analyse the radiative energy available for melt at the surface over the GrIS in summer and autumn (sensible- and latent heat flux have also been studied in Figure S4 in supplementary, but shows to be insignificant). We observe a similar general behaviour of the radiative SEB components (i.e. net shortwave – $SW_{net}$, downwelling shortwave – $SWD$, net longwave – $LW_{net}$, downwelling

longwave – *LWD* and upwelling longwave – *LWU*) with warming temperatures from MAR CMIP5 and MAR CMIP6 over both summer (JJA, Figure 2 a) and autumn (SON, Figure 2 b). Here, the $SW_{net}$ (yellow) reaching the surface is increasing with warming – an effect that seem to be coming from the melt-alebdo feedback and darkening of the surface – despite decreasing SWD (purple), concurrently as LWD (red) is increasing. However, while we observe similar behaviour of the projection of SEB components between MAR CMIP5 and MAR CMIP6 in autumn (SON, Figure 2 b), differences are found between the two ensembles in summer (JJA, Figure 2 a).

Although the net radiative flux (grey) is behaving similarly for MAR CMIP5 and MAR CMIP6 for a given near-surface temperature increase in summer (JJA, Figure 2 a), the shortwave and longwave components composing the net radiative flux show different behaviour between the two ensembles for this season. We observe that there is more $SW_{net}$ radiation reaching the surface in MAR CMIP6 compared to MAR CMIP5 for a given warming, which mainly come from more SWD reaching the surface in MAR CMIP6. Further, we observe less net longwave flux ($LW_{net}$, blue) absorbed at the surface in MAR CMIP6 compared to MAR CMIP5. The amount of outgoing longwave radiation emitted from the surface (LWU, green) is behaving similarly between the two ensembles, so the difference we observe in $LW_{net}$ comes from a decrease in downwelling longwave radiation (LWD, red) for a given temperature increase in MAR CMIP6.

The difference in summer LWD and SWD is indicative of an effect from a change in cloud properties (i.e. cloud cover and cloud optical depth (COD)), affecting the transmissivity and emissivity of the atmosphere. In terms of cloud cover, MAR CMIP5 and MAR CMIP6 show a diverging behaviour over both seasons (Figure 2 c and d). While the magnitude of the changes in cloud cover in summer are the same between MAR CMIP5 and MAR CMIP6 with warming, we see completely different behaviour in terms of the sign of the change (Figure 2 c). In MAR CMIP5 cloud cover increases (+2.4 $\pm$ 0.9 %) with warming (+5.4°C), while in MAR CMIP6 cloud cover over the GrIS notably decreases (-2.2$\pm$ 1.1% ) for a given warming (+5.4°C, Figure 2 c, see Table T1 in supplementary). The decrease in cloud cover explains why we also see more SWD reaching the surface in summer in MAR CMIP6 compared to MAR CMIP5 (Figure 2 a) as the transmissivity of the atmosphere increases with decreasing cloud cover. Moreover, the magnitude of the difference is not as pronounced in autumn (SON) as for summer (JJA). For autumn, there is an increase in cloud cover of +2.9 $\pm$ 0.8 % in MAR CMIP5 and a decrease of -0.7 $\pm$ 0.9 % in MAR CMIP6 for a +6.7°C near-surface warming (Figure 2 d, see Table T1 in supplementary).

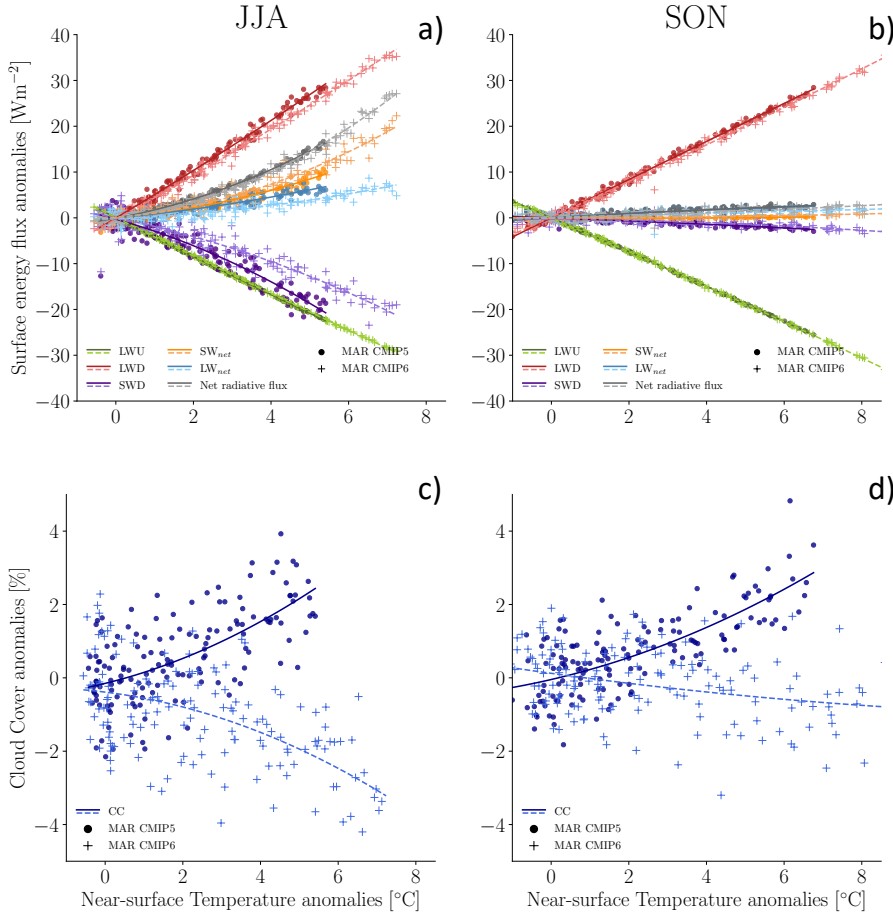

**Figure 2. Radiative SEB component anomalies [Wm$^{-2}$] and total cloud cover anomalies [%] as a function of the near-surface air temperature anomalies [°C]. a)** Seasonal (JJA) radiative SEB component anomalies [Wm$^{-2}$] over the GrIS according to near-surface air temperature anomalies [°C] from MAR CMIP5 (dots) and MAR CMIP6 (crosses), with regression drawn in solid lines for MAR CMIP5 and dashed lines for MAR CMIP6. It includes the radiative energy fluxes of longwave down (LWD), longwave up (LWU), shortwave down (SWD), net longwave radiation (LW$_{net}$), net shortwave radiation (SW$_{net}$), and Net radiative flux. Positive direction towards the surface. **b)** same as **a)**, but for autumn (SON). **c)** same as **a)**, but for the variable of cloud cover [%]. **d)** same as **c)**, but for autumn (SON).

We suggest the higher amount of SWD reaching the surface of the GrIS in MAR CMIP6 is primarily a consequence of the decrease in cloud cover with warming during summer (JJA). Because we do not see major differences in cloud optical depth (Figure 3 a), we can likely rule out a notable contribution from changes in cloud microphysics between MAR CMIP5 and
205 MAR CMIP6 over Greenland (i.e. water phase changes) in summer.

Conversely, in autumn (SON, Figure 3 b) there is a small difference in COD anomaly (Figure 3 b). We see an increase of + $0.8 \pm 0.01$ in MAR CMIP5 and + $0.1 \pm 0.01$ in MAR CMIP6 for a + 6.7 °C near-surface temperature warming (see Table T1

in supplementary). For autumn (SON), the data therefore suggests that optically thicker clouds counteract the effect of cloud cover reduction in MAR CMIP6 (Figure 2d). Therefore, we observe no difference in the autumn SW and LW radiative energy (Figure 2b) between MAR CMIP5 and MAR CMIP6 than can explain the difference in the autumn SMB (Figure 1c).

We have also studies the radiative surface energy flux and cloud cover change for the GrIS ablation zone and accumulation zone individually. We provide methods and results in supplementary Figure S17 and S18.

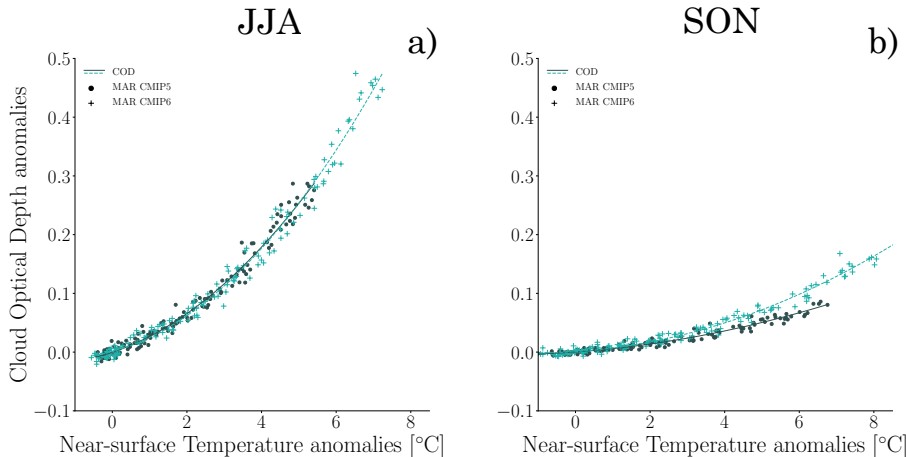

**Figure 3. Cloud Optical Depth anomalies over the GrIS according to near-surface temperature anomalies [$^{\circ}$C]. a)** Summer (JJA) Cloud Optical Depth (COD) anomalies over the GrIS as a function of annual near-surface temperature anomalies [$^{\circ}$C] from MAR CMIP5 (dots) and MAR CMIP6 (crosses), with regression drawn in solid lines for MAR CMIP5 and scattered lines for MAR CMIP6. All anomalies are related to the thirty-year average reference period (1961–1990). **b)** same as **a)** but for autumn (SON).

## 3.3 Spatial distribution over the GrIS

We have observed how the spatially averaged GrIS surface mass and energy balance components change on long time scales.
However, as the spatial distribution of the SEB, and therefore also the SMB, over the GrIS is not uniform we further explore the
spatial change in cloud cover, its effect on the radiative budget at the surface, and finally the SMB response, for a twenty-year
warming period (+ 4 $^{\circ}$C $\pm$ 10 years) compared to a thirty-year averaged reference period (1960-1990).

### 3.3.1 Cloud cover

The different cloud cover responses over the GrIS in summer (Figure 4, JJA) project spatially homogeneous patterns. While
we observe an overall increase in total cloud cover in MAR CMIP5 with warming over most of the GrIS in summer (JJA), we
see a general decrease in MAR CMIP6. Due to the homogeneous nature of the difference between MAR CMIP6 and MAR
CMIP5 (JJA, "(CMIP6-CMIP5)") we argue that differences in circulation are unlikely to be the driver behind this difference
in cloud cover response with warming, as circulation-driven cloud cover change would be expected to result in a more distinct
spatial pattern in areas with anomalous upslope and downslope winds. Additionally, the data also suggests that this pattern in
cloud cover stems mostly from a contrasting behaviour in upper-level clouds ( < 440 hPa), while mid-level clouds ($\geq$ 440 hPa,

$\le$ 680 hPa) and low-level clouds ( $>$ 680 hPa) do not behave very differently (see Figure S3 in supplementary for JJA mid- and low-level clouds). We expect this homogeneous pattern of cloud cover reduction in MAR CMIP6 to lead to a greater proportion of shortwave radiation reaching large parts of the surface of the GrIS in summer in MAR CMIP6.

In autumn we detect similar patterns of cloud cover trends with warming between MAR CMIP5 and MAR CMIP6 (Figure 4, SON), but of different magnitude. In the total cloud cover, MAR CMIP5 shows increasing cloud cover over most regions of the ice sheet, with an exception of modest decrease over the extreme south-east. In the north-west area where MAR CMIP5 shows its strongest positive cloud cover anomaly, MAR CMIP6 only shows a modest increase, whereas the rest of the ice sheet experiences a decrease in cloud cover. We also see more negative cloud cover anomaly over the whole ice sheet in MAR CMIP6 when compared to MAR CMIP5 (SON, "(CMIP6-CMIP5)"). Again, the data also suggests for the autumn clouds that this pattern in cloud cover stems mostly from a contrasting behaviour in upper-level clouds, while mid- and low-level clouds do not behave very differently (see Figure S4 in supplementary for SON mid- and low-level clouds).

In Figure S13 and S14 in supplementary we show in detail the cloud cover response for each of the six CMIP5 models and five CMIP6 models respectively, where the individual models (except MIROC5) generally captures the ensemble mean well. Therefore, we argue that the models chosen for downscaling capture the overall cloud cover response for CMIP5 and CMIP6.

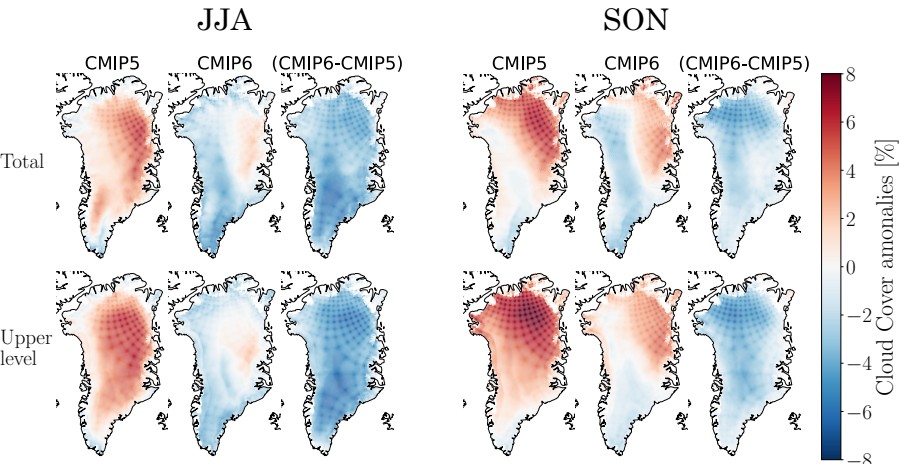

**Figure 4. Cloud cover anomalies [%] for MAR CMIP5 and MAR CMIP6 simulations (+ 4 °C ± 10 years) for summer (JJA) and autumn (SON).** Twenty-year average (4 °C ± 10 years) of the cloud cover [%] over the GrIS for summer (JJA, left) and autumn (SON, right). The two rows indicate the total (upper row) and upper level cloud cover ( < 440 hPa, bottom row). Each season has three columns — the first indicates the cloud cover anomalies for MAR CMIP5, the second for MAR CMIP6, and the third the difference between the two (CMIP6-CMIP5). For MAR CMIP5 and MAR CMIP6 a positive value (red) indicates an increase in cloud cover, and a negative value (blue) a reduction in cloud cover compared to the reference period. For the difference (CMIP6-CMIP5) a positive value (red) indicates more positive cloud cover anomaly, and negative values (blue) indicate a more negative cloud cover anomaly in MAR CMIP6 compared to MAR CMIP5.

### 3.3.2 Radiative Surface Energy Budget

Figure 5 shows the difference between MAR CMIP6 anomaly and MAR CMIP5 anomaly of the radiative SEB components for summer (JJA, left) and autumn (SON, right) (See Figure S5 and S6 in supplementary for individual MAR CMIP5 and MAR CMIP6 model mean anomaly for JJA and SON respectively). Hereafter, a lower anomaly means that there is a more negative MAR CMIP6 anomaly compared to the MAR CMIP5 anomaly, and similarly a higher anomaly refer to a more positive MAR CMIP6 anomaly compared to MAR CMIP5 anomaly.

In summer (Figure 5, JJA), there is a lower downwelling LW radiative flux (LWD) anomaly, and a higher downwelling SW radiative flux (SWD) anomaly over most of the ice sheet. This corresponds to what we expect from the homogeneous cloud cover reduction in summer in MAR CMIP6, increasing the transmissivity of the atmosphere. Moreover, there is overall lower LW$_{net}$ radiation anomaly over the GrIS. With smaller amounts of LWD radiative flux over the GrIS from a reduction of cloud cover, we expect there to be less heat trapped, hence less warming of the snowpack from LW radiation. Further, because there was an overall increase in SWD radiation over the ice sheet, we have a higher SW$_{net}$ radiation anomaly in MAR CMIP6 concentrated along the edges where we find the dark ablation zone. We also see an interesting band of slightly lower SW$_{net}$ anomaly just above the ablation zone. We suspect this pattern to stem from the influence of reduced cloud cover over

an area where clouds usually warm the high-reflective surface, i.e. less clouds cool the surface. Therefore, due to a reduction in clouds in MAR CMIP6 the net energy flux is reduced above the ablation zone, leading to less melt over the bright surface and therefore a less pronounced melt-albedo feedback, causing a reduction in absorbed shortwave radiation in MAR CMIP6. Over all of the accumulation zone the snowpack will usually reflect more of the incoming SW radiation and a reduction in cloud cover therefore leads to a cooling of the snowpack as a result of less $LW_{net}$ radiation. Conversely, in the darker ablation zone where more bare ice is exposed, more of this extra SWD radiation in MAR CMIP6 can be absorbed and lead to more melt or warming of the surface.

In autumn (Figure 5, SON), there is a lower LWD anomaly over most of the GrIS, with an exception along the east coast were the anomaly is modestly higher. As for summer (JJA), there is an overall lower $LW_{net}$ radiation anomaly across the whole ice sheet, however of smaller magnitude. A modestly higher SWD anomaly is also seen across most of the ice sheet, however slightly less in the extreme south-east. Therefore, we suggest the higher $SW_{net}$ radiation anomalies – mostly centred around the southern ablation zone – to come from a darker surface created in summer and still being exposed over the lower ablation zone during autumn. We expect this change in SW radiation around the lower ablation zone to enhance autumn melt and runoff over the lower ablation zone.

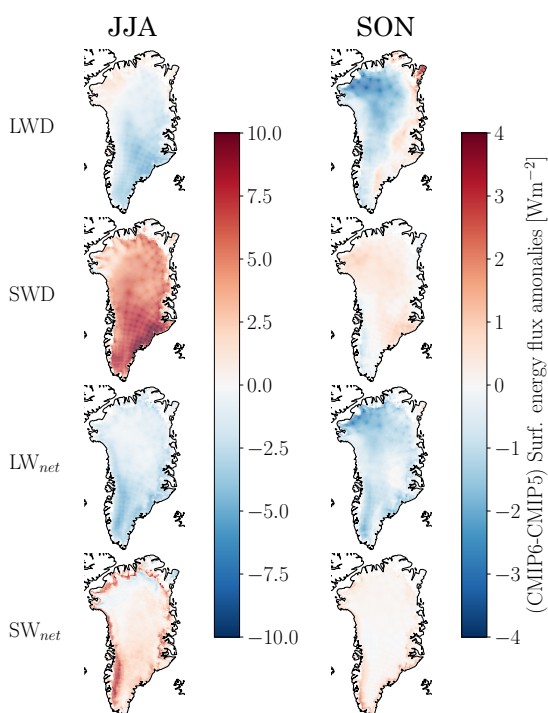

**Figure 5. Difference in anomaly of SEB fluxes between MAR CMIP6 and MAR CMIP5 simulations (+ 4°C ± 10 years) for summer (JJA) and autumn (SON).** Twenty-year average (+ 4°C ± 10 years) difference in SEB. The two columns indicate the mean difference of the anomalies for summer (JJA) (left) and autumn (SON) (right). Positive values (red colours) indicate a greater energy flux reaching the surface in MAR CMIP5 compared to MAR CMIP6, whereas negative values (blue colours) indicate a smaller energy flux reaching the surface in MAR CMIP6 compared to MAR CMIP5. **Note:** the colourbars in summer (JJA) and autumn (SON) are not on the same range. See Figure S15 in supplementary for relative change plots.

### 3.3.3 Surface Mass Balance response

Figure 6 shows the difference between MAR CMIP6 anomaly and MAR CMIP5 anomaly of SMB, melt, runoff and refreezing
for summer (JJA, left) and autumn (SON, right) (See Figure S7 and S8 in supplementary for individual MAR CMIP5 and MAR CMIP6 model mean anomaly for JJA and SON respectively). Again, a lower anomaly means that there is a more negative MAR CMIP6 anomaly compared to the MAR CMIP5 anomaly, and similarly a higher anomaly refer to a more positive MAR CMIP6 anomaly compared to MAR CMIP5 anomaly.

In summer (JJA), there is a lower SMB anomaly over most of the ablation zone, and a higher SMB anomaly in the transition
zone between the ablation zone and the accumulation zone (i.e. the percolation zone). The more negative SMB in MAR CMIP6 (relative to MAR CMIP5) appear to be coming from a higher melt and runoff anomaly in the same area, most likely induced by more absorbed SW radiation. Conversely, in the percolation zone we suspect an effect of higher refreezing anomaly

and subsequent lower runoff anomaly in MAR CMIP6, yielding the higher SMB anomaly for this area. Most likely, despite enhanced SW radiation due to less clouds, the simultaneous decreased incoming LW radiation reduces the bare ice exposure in the percolation zone where the residual snowpack is usually thicker and reflects sunlight more efficiently for longer. Thus, the ice sheet experiences less melt and runoff, and more refreezing in this area during summer (JJA). The effect of lower SMB anomaly in the ablation zone looks to be cancelling the effect of higher SMB anomaly over the percolation zone, thus no difference is detected between MAR CMIP6 and MAR CMIP5 in the spatially averaged SMB projection with warming in summer (Figure 1 b).

In autumn (SON) we do not see the same buffering effect of more refreezing in MAR CMIP6 in the percolation zone as we saw for summer (JJA), partly due to a decrease in meltwater production in this region. Figure 6 shows lower SMB anomaly mostly over the ablation zone, concentrated around the southern tip of the ice sheet. Over the lower parts of the ablation area in the south of the ice sheet we also see higher melt and runoff anomalies. We believe that the excess SWD reaching the surface in MAR CMIP6 in summer (JJA) induces a stronger surface darkening of the lower ablation zone, thus enhancing the surface mass loss in autumn (SON) where the bare ice is still exposed. With a reduced buffering effect of refreezing in the percolation zone, compared to summer (JJA), we then detect a total difference in the spatially averaged autumn SMB from more melt and runoff in the darker ablation zone.

Parts of the refreezing in the percolation zone in MAR CMIP6 can possibly be explained by the faster warming pace of the CMIP6 ensemble (not shown). Different warming rates imply that the selection of the twenty-year period of $\sim 4°C$ warming (see Table T2 in supplementary), 'faster warming' models have not had the same time to fully extend the ablation zone to higher elevations to the same extent as the other models that reach the same atmospheric temperature at a later stage. In the 'slower warming' models the ablation zone is more in equilibrium with the warming climate and therefore likely already larger in CMIP5 than in CMIP6 at a given warming, because CMIP6 generally warms faster. This could explain parts of the higher refreezing in the percolation zone, because CMIP5 might already have gotten rid of the buffering snow layer.

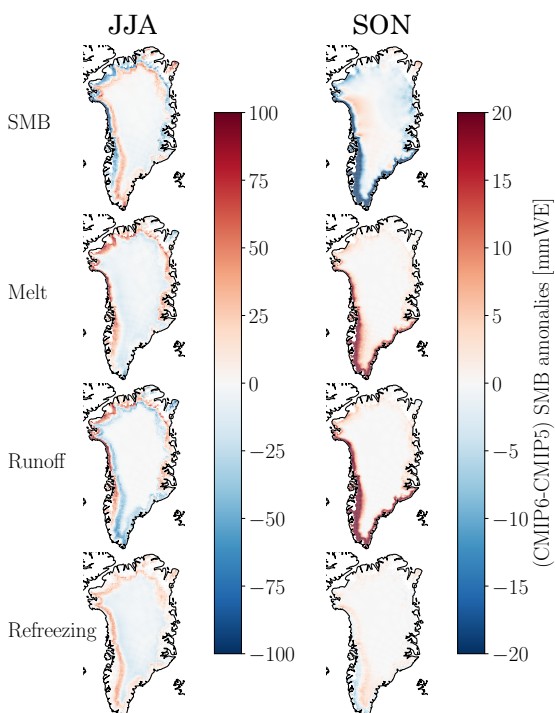

**Figure 6. Difference in anomaly of selected SMB components between MAR CMIP6 and MAR CMIP5 simulations (+ 4°C ± 10 years) for summer (JJA) and autumn (SON).** Twenty-year average (+ 4°C ± 10 years) difference in anomalies of melt, runoff, refreezing and the total SMB [mmWE], of MAR (CMIP6-CMIP5). Anomalies are related to the reference period (1961–1990). The two columns indicate the mean difference of the anomalies for summer season (JJA, left), and autumn season (SON, right). Positive values (red) indicate a greater mass balance at the surface in MAR CMIP6 compared to MAR CMIP5, while negative values (blue) indicate a lower mass balance in MAR CMIP6 compared to MAR CMIP5. **Note:** the colourbars in summer (JJA) and autumn (SON) are not on the same range. See Figure S16 in supplementary for relative change plots.

### 3.4 Melt-albedo feedback response

Future changes in the strength of the melt-albedo feedback plays a leading part in how much energy is available for melt over the GrIS surface. We detect a decrease in the spatially averaged albedo anomaly as a function of increasing seasonal near-surface temperature for both MAR CMIP5 and MAR CMIP6 in summer (Figure 7 a), the well-known melt-albedo feedback. However, there is no difference in the spatially averaged albedo anomalies between MAR CMIP5 and MAR CMIP6, but the differences in spatial summer (JJA) distribution between MAR CMIP5 and MAR CMIP6 (Figure 7 c) show a lower albedo anomaly in the ablation zone around the ice sheet for MAR CMIP6, with the exception of the south-east coast. We believe this darkening of the surface is due to more SW radiative flux over the ablation zone of bare ice, induced from decreasing

cloud cover in MAR CMIP6. This surplus of SWD radiation in MAR CMIP6 leads to more melt over the dark ablation zone, enhancing surface darkening.

Conversely, we observe more positive anomalies over parts of the percolation zone in MAR CMIP6 compared to MAR CMIP5 (See Figure S9 and S10 in supplementary for individual MAR CMIP5 and MAR CMIP6 model mean anomaly for JJA and SON respectively). We suspect that parts of the percolation zone in MAR CMIP6 experience a surface cooling, as a result of less LW radiation reaching the surface, stemming from the reduction in cloud cover. Therefore, this layer has a higher albedo because there is winter snow in this area that has experienced less long-lasting melt events in MAR CMIP6. We suggest

that the more negative albedo anomaly detected in the ablation zone cancels out the more positive albedo in the percolation zone, thus there is no difference in the spatially averaged projection of the albedo between MAR CMIP6 and MAR CMIP5 in summer (JJA).

    A general decrease in albedo anomalies with increasing near-surface temperature is also detected for both MAR CMIP5 and MAR CMIP6 in autumn (Figure 7 b). Here, MAR CMIP5 projects a decrease in albedo of -0.008 ±0.001 and MAR CMIP6

a decrease of -0.014 ±0.001 for a temperature increase of +6.7°C (see Table T1 in supplementary). The spatial distribution reveals a more negative albedo anomaly in MAR CMIP6 compared to MAR CMIP5 around the outer ablation zone (Figure 7 d). This lower albedo in autumn in MAR CMIP6 is likely due to the fact that the higher ablation zone gets covered with snow during autumn (SON) first. However, the lower ablation zone experiences more melt and runoff in summer due to less clouds, and therefore is darker coming into autumn. Here, the bare ice is still exposed in the lower ablation zone, leading to more

absorption of SW radiation and surface melt in the autumn (SON), without a buffering effect from extra refreezing that we observed in summer (JJA).

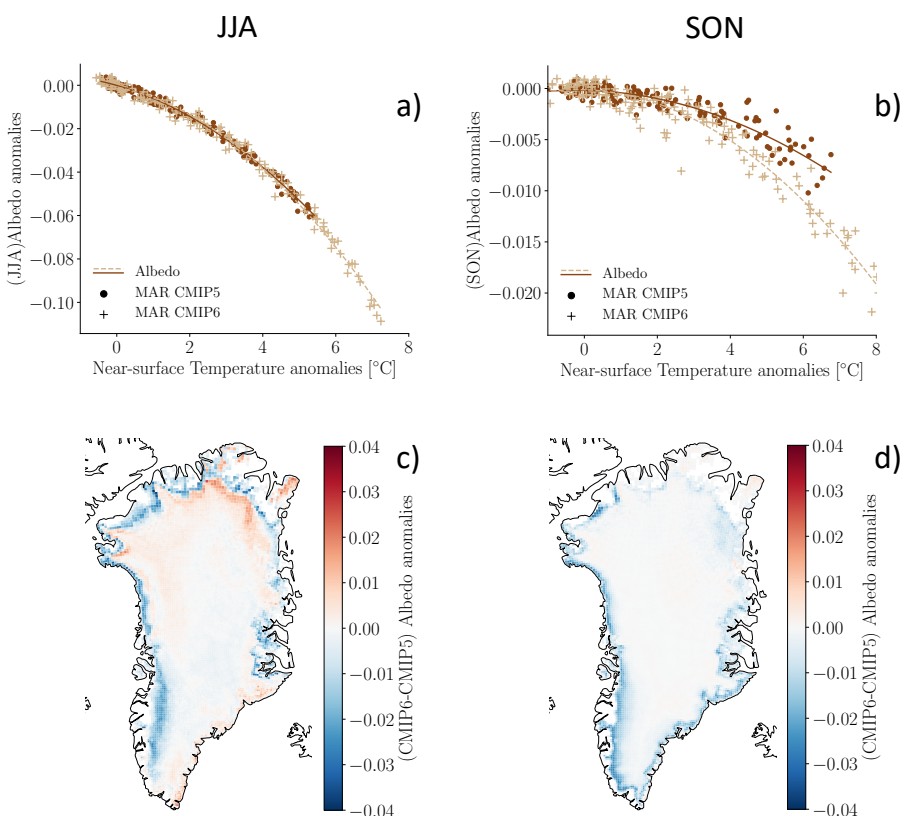

**Figure 7. Albedo anomalies as a function of near-surface air temperature anomalies [°C] over GrIS and spatial distribution for a warming period of 4°C**. Seasonal albedo anomalies according to seasonal near-surface temperature [°C] (top), and spatially distributed difference in change over GrIS for 4°C warming (bottom) for summer season (JJA) (left) and autumn season (SON) (right). For the spatial maps, areas of positive values (red colours) indicate higher values of albedo, and negative values indicate lower values of albedo in MAR CMIP6 when compared to MAR CMIP5.

## 4  Discussion and conclusion

In this study, we performed an analysis of high-resolution regional climate model simulations over the GrIS to investigate possible physical mechanisms driving the excess SMB loss in CMIP6 models. Previous to this study, it was believed that the excess SMB reduction seen in CMIP6 compared to CMIP5 was solely a product of a greater Arctic amplification signal in CMIP6 models. Our work suggests that parts of the greater mass loss in CMIP6 over the GrIS is driven by a difference in SMB sensitivity, from a change in cloud representation in CMIP6 models.

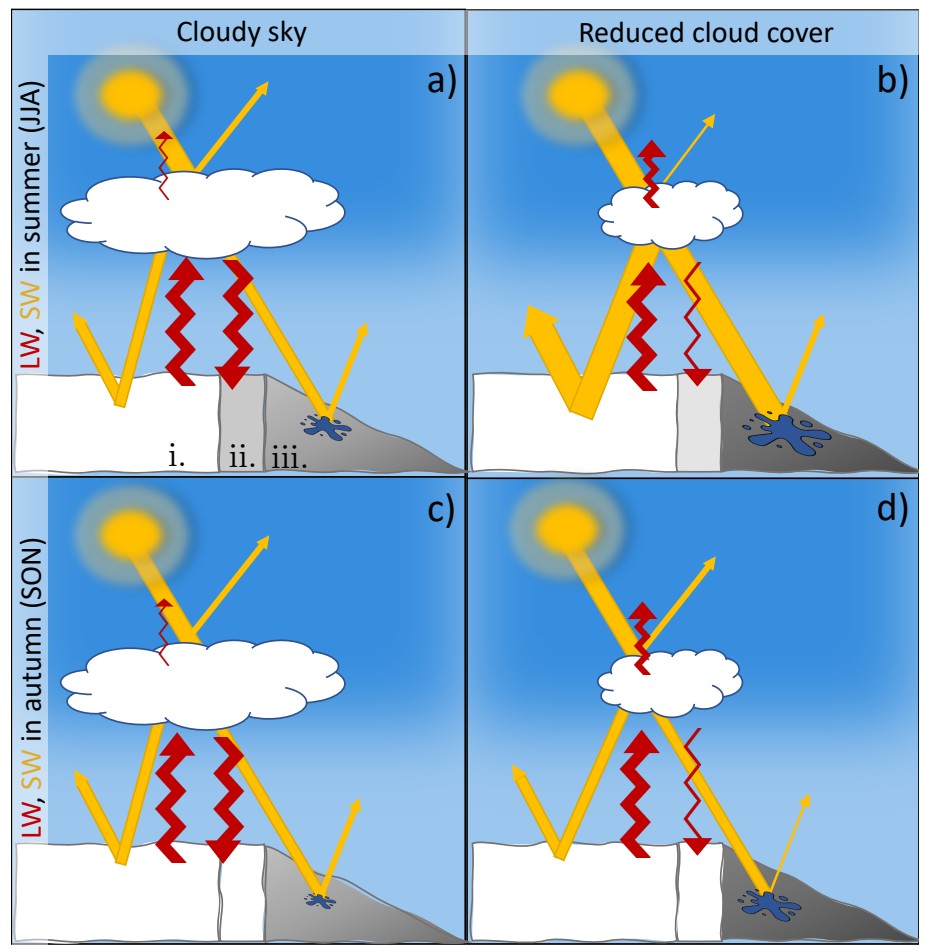

**Figure 8. Schematic radiative energy flows over the GrIS in CMIP5 'Cloudy sky' and CMIP6 'Reduced cloud cover'.** Simplified schematic representation of the radiative energy flows over the Greenland ice sheet for cloudy sky condition (left pannel) and reduced cloud condition (right panel), for summer (JJA, a-b) and autumn (SON, c-d). All fluxes are positive downwards. The ice sheet is divided into three different zones i) the accumulation zone, ii) the percolation zone, iii) the ablation zone.

By comparing two model ensembles of six CMIP5 RCP8.5 and five CMIP6 SSP5-8.5 future projections for a given temperature increase, we found a greater sensitivity of Greenland surface mass loss in CMIP6 centered around summer and autumn. Yet, the difference in mass loss between CMIP5 and CMIP6 was largest during autumn.

Assessment of future changes in the SEB and cloud properties over the GrIS suggested a reduction in high clouds during summer and autumn to be the main driver of additional SW radiation reaching the surface in CMIP6, while cloud cover increases with warming in CMIP5. However, the detailed mechanisms behind different cloud cover trends with warming in CMIP6 compared to CMIP5 are still unanswered. We suggest that future studies look in more detail to the circulation in both CMIP5 and CMIP6 over Greenland, although initial analysis suggest no notable difference between the two ensembles (Delhasse et al., 2020).

The impact of a reduction in cloud cover on the radiative energy budget and the SMB over the GrIS is schematically represented in Figure 8. Our data showed a stronger melt-albedo feedback in CMIP6 mainly in the lower ablation zone where bare ice is continuously exposed during summer and early autumn. Here we observed more surface melt and darkening of the surface during summer, from an increase in SWD from reduced cloud cover (Figure 8 a-b). However, because of a competing mechanism in the percolation zone leading to more refreezing (Figure 8 a-b), there was no difference in the total SMB projection between CMIP5 and CMIP6 in summer. In the autumn however, there was a stronger melt and runoff signal in CMIP6 from the darkening of the lower albedo zone in summer (Figure 8 c-d).

In summary, our analysis highlights that Greenland is losing more mass in CMIP6 due to two factors;

1) a (known) greater sensitivity to GHG emissions and therefore warmer temperatures

2) previously unnotified cloud-related surface energy budget changes between CMIP5 and CMIP6 that enhance the GrIS sensitivity to warming.

*Code and data availability.* We provide examples of the Python code used to analyse the MAR model results. The code can be found at https://github.com/idunnam/MAR_CMIP5_CMIP6_analysis.git. All the MAR model results are available for download on ftp://ftp.climato. be/fettweis/MARv3.9/ISMIP6/GrIS/ in the framework of the ISMIP6 exercise (https://tc.copernicus.org/articles/14/2331/2020/). CMIP5 and CMIP6 model outputs can be openly accessed via different ESGF data nodes (e.g., https://esgf-node.llnl.gov/projects/cmip6/, https://esgf-node.llnl.gov/projects/cmip5/).

## Appendix A

### A1

*Author contributions.* SH designed the study with contribution from IAM and TS. XF ran the MARv3.9 simulations and provided the outputs. IAM analysed all data, produced the figures, and prepared the manuscript . All authors discussed the results and were involved in editing the manuscript.

*Competing interests.* We do not have any competing interests.

*Acknowledgements.* This project has received funding from the European Research Council (ERC) under the European Union's Horizon
2020 research and innovation program (Grant agreement number 758005). The computations and simulations were performed on resources provided by UNINETT Sigma2 – the National Infrastructure for High Performance Computing and Data Storage in Norway.

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
