# Peer review of "Cloud- and ice-albedo feedbacks drive greater Greenland ice sheet sensitivity to warming in CMIP6 than in CMIP5"

_The Cryosphere, 2023_

## Author Comment (AC1)

**Answer to Referee #1 – Manuscript tc-2023-24**

Mostue and co-authors explore summer and autumn differences in surface mass and energy budget (SMB and SEB) components as well as cloud cover based on output of a regional climate model forced by several global climate models from two generations under the high-end baseline scenario. They start by relating anomalies in near-surface temperature with individual SMB and SEB components, showing similar behavior and magnitude to SMB, melt, runoff, longwave down and longwave up. However, the set of models used from the most recent generation of global climate models projects more warming than previously. Finally, it is shown a contrasting relationship between the anomaly of near-surface temperature and cloud cover anomaly between the two generations of global climate models. The authors hypothesize that the cloud cover decrease allows more absorption of solar radiation by the surface, generating enhanced surface melt and runoff in the ablation zone in summer extending to autumn.

We would like to thank Referee #1 for taking the time to read our manuscript so thoroughly and provide in-depth feedback on our study. Below we have responded to their comments and how we would like to address them. We think that our manuscript will greatly benefit from the adjustments.

This piece of work explores relevant scientific aspects, but the methods and set of variables used are not sufficient to prove the robustness of the results. The level of detail provided in the Section 3.1 and 3.2 is commendable, but presents an unnecessary detailed picture between near-surface temperature with SMB and SEB components. The most relevant part starts with the lower panels in Figure 2, serving as motivation for the rest of the manuscript. Even though the authors show no relevant changes in cloud optical thickness, it would be worthwhile to explore changes in cloud microphysics and its relationship with cloud cover. In addition, summer and autumn precipitation should be included in the analysis, given its role to surface albedo.

My comments are provided by line number (LN) or specific figure below.

First of all thank you for highlighting that our manuscript 'explores relevant scientific aspects'. We have highlighted below how we will adjust our manuscript relating to more specific comments.

However, we do not agree with Reviewer #1 that Sections 3.1 and 3.2 contain too much detail when connecting the surface mass balance and surface energy budget projections over Greenland to the near-surface temperature anomaly. These results present some of the most novel findings. One of the main issues highlighted by the latest IPCC AR6 report is that the CMIP6 model mean or related impact studies (as presented here) can not be used at face value anymore, because some models have unrealistic temperature sensitivities compared to proxy data. The only way around that issue is to compare the surface mass balance and surface energy budget projections at a given temperature anomaly - something that has not been done for any of the two large ice sheets, Greenland and Antarctica. In addition, the literature about Greenland SMB and SEB projections is very sparse to begin with, and nobody has studied it for CMIP6 in great detail, let alone compared CMIP5 to CMIP6 high-resolution climate projections. While we agree that the level of detail might be too great for some parts of the scientific community more interested in the general magnitude of sea level rise, we still believe that there will be interest in our results.

Regarding the comment "The most relevant part starts with the lower panels in Figure 2..":

While Figure 2 (lower panel), showing cloud cover changes vs Greenland temperature

anomaly for summer and autumn, might be one of the most interesting findings of the paper and could deserve its own publication, we would like to highlight that we have chosen to focus on the connection between SMB and all of the most important physical drivers of the differences between CMIP5 and CMIP6 SMB projections in this manuscript. Solely focusing on clouds and their microphysics is not the scope of this study, but would certainly present an interesting additional publication (the data of the presented simulations is open-source and therefore open to anyone who would like to perform this analysis).

The Introduction is short and does not summarize/highlight what the scientific community has recently done concerning the impact of clouds and surface albedo on SEB and SMB components over the Greenland ice sheet.
We agree that the 'Introduction' could be expanded with respect to the latest literature on the impacts of clouds, albedo, the SEB and SMB components. We have expanded our introduction accordingly in the revised manuscript.

 LN22: the accelerating mass loss pace since the mid-1990s is not only a consequence of increased temperatures from anthropogenic greenhouse gases, but rather a consequence of a superimposed effect with extraordinary atmospheric conditions in recent summers (Bennartz et al. 2013, Fausto et al. 2012, Tedesco et al. 2011, Tedesco et al. 2016).
We agree and have rephrased and expanded this part of the 'Introduction' to highlight the recent anomalous phase of the North Atlantic Oscillation and the connected increase in high-pressure phases of the Greenland Ice Sheet during the summer months (JJA) (Bennartz et al. 2013, Fausto et al. 2012, Tedesco et al. 2011, Tedesco et al. 2016).

LN28: the SMB definition should not be part of the Introduction, but in the Methods section, naming individual components and explaining how do you define accumulation and ablation zones.
We agree that the SMB definition could be moved to the 'Methods'. We have changed this accordingly in the revised manuscript.

LN31: in addition to solar radiation, consider the role of sensible heat flux to darken the surface (Wang et al. 2021)
Sensible heat flux is mostly relevant very close to the tundra (i.e. where the ice sheet ends). We have studied the Sensible heat flux (and Latent heat flux) and they are insignificant in extreme-high-emission scenarios over Greenland when comparing CMIP5 and CMIP6 MAR projections. We have included this information in the revised manuscript and added the plots for these terms in the Supplementary Material S.4.

LN41: the authors should address the fact that as a consequence of more open waters, CMIP6 projects more precipitation and more rainfall in Greenland than CMIP5 (McCrystall et al. 2021). This point can also be later discussed as a factor contributing to decreasing albedo, as also shown by Box et al. (2022).
We agree that more open water and changes in precipitation certainly contribute to the difference between our CMIP5 and CMIP6. Both of these factors have been discussed in more detail in Hofer et al. (2020) using the same simulations as here. Rainfall projections only differ from 2070 onwards between CMIP5 and CMIP6 MAR simulations, however, the SMB and melt projections start to diverge already from 2020 onwards. We therefore do not think that extra rainfall is the main driver behind the diverging SMB sensitivities, albeit it might still be a contributing factor later on in our projections. We have now added references to McCrystall et al (2021) and Box et al. (2022) and a few more sentences highlighting this discussion in the revised manuscript.

LN50: state that a surplus in SEB is energy available for melt and not necessarily surface melt

This is a semantic discussion about what constitutes the "surface". Most of the radiative and non-radiative energy fluxes are absorbed within a few cm of the first ice/snow crystals of the surface, and surface-energy-budget-triggered melt usually melts ice/snow crystals within the first few centimeters. Therefore, we do not see what the reviewer means when they distinguish between "melt" and "surface melt" in connection to the surface energy budget.

LN64: the last paragraph of Section 2.1 could be moved to the Introduction, where a few of these references could better distilled

This was done accordingly in the revised manuscript.

LN72: it would be relevant to explain here why only RCP8.5 and SS5-8.5 is chosen for the study, as Hofer et al. (2020) made use of all the projected scenarios

Hofer et al. (2020) also only used MAR simulations for the extreme-high-emission scenarios RCP8.5 and SSP5-8.5. The authors then used statistical extrapolation combined with the outputs of the actual CMIP5 and CMIP6 GCMs to infer their surface mass balance projections for other emission scenarios.

LN75: it is also unclear why the period 1961-1990 is chosen. I would assume the last 3 decades (1991-2020), responsible for the accelerated mass loss, a better period for comparison with future projections

We chose to use the period 1961-1990 as our thirty-year average reference period because the GrIS was assumed to be in a stable state (van den Broeke et.al. 2016).

Calculating the mean only really reflects the general distribution when the values for that period roughly follow a normal distribution. Therefore, choosing the mean over the indicated period of 1991-2020, which exhibits a clear linear trend, does not make too much sense in our opinion.

We have added more detail to that part of the manuscript explaining the choice of our reference period.

LN82: it should be indicated how the ice cover mask (more than 10\% ice cover) can influence the following results

With a 10% ice cover mask that does not change over time we expect our SMB reduction to be slightly overestimated compared to a dynamic ice mask, but recent research indicated this error to be somewhere between 1% and 6 % (Kjeldsen et.al. 2020, Hansen et al.2022).

We have expanded on that part of the manuscript considering this and added the corresponding references.

LN83: it is unclear why a twenty-averaged period for ~4ºC is chosen for the dissemination of certain the results
LN85 how can you gain insight of changes caused by rapid Greenland warming using a twenty-year averaged period?

Throughout the manuscript we use the full time-range available from our simulations, which

is until 2100 (e.g. Fig.1 and 2). We have chosen a +4C threshold (+-10 years) as a specific focus to be able to compare all models for the same temperature increase. The individual CMIP models warm at different rates, thus do not reach the same temperature by the end of the century. +4C is the highest temperature rise for a twenty-year-averaged period where we have data for all CMIP5 and CMIP6 models, therefore the reason for this choice.

We have added more detail to that part of the manuscript explaining the choice of this warming period.

Figure 4, 5, 6 and 7: use statistical inference to indicate the level of confidence in changes between CMIP5 and CMIP6.
LN115: could you present the same charts (Figure 1 and 2) but for the differences between CMIP5 and CMIP6, making use of statistical inference to state the robustness of the mentioned differences?
For the variables in Figure 1 and 2 (i.e. SMB, ME, RU, SEB components, and Cloud Cover) we have calculated the R2 score (Mostue 2022, Table B.2) and the one standard deviation around the regression lines (Supplementary Material Table T1).  Where we state that there is a difference between CMIP5 and CMIP6 in the manuscript, we find the difference between the regression lines to be larger than the one standard deviation and therefore statistically significant.

Figure 1: legends and axis labels missing. Also, consider making the season as a subtitle of the subplot as in Figure 2
Thank you for pointing this out. This happened during the processing of our file in the TC submission system. We will ensure the labels are there in the revised manuscript.

LN139: start the sentence with "In SON" instead of "Here". Otherwise, it is not clear to which season this sentence belongs
As we state in LN136-139 we are commenting on the general trend of the SEB components for both seasons. Therefore in our opinion, 'Here' is appropriate in LN139, and should not be season specific.

LN147: in LN139 you explain that more SW$_{net}$ is due to darkening and here is due to SWD. Please, rephrase it.
In LN139 we explain the reason for the general increasing trend of SWnet for a given temperature increase for both seasons and both CMIP5 and CMIP6 individually. Then, in LN147 we consider the differences in SWnet between CMIP5 and CMIP6 (only seen in summer), and explain it by the difference in SWD.

We have slightly adapted the wording of these two sentences to avoid any confusion for the readers.

Figure 2: legends missing and temperature unit incomplete
Thank you for pointing this out. This happened during the processing of our file in the TC submission system. We will ensure the labels are there in the revised manuscript.

LN165: why do you assume that no differences in cloud optical depth means no differences in cloud microphysics? Isn't this statement contradicting Hofer et al. (2019)? Could you elaborate your thought?
In LN165 we talk about the reason for higher incoming solar radiation reaching the surface in CMIP6 simulations. Because we cannot find any difference in cloud optical depth it is only reasonable to assume that most of the extra solar radiation is coming from the decrease in cloud cover. Given that cloud optical depth for a given total cloud water content is defined by cloud microphysics (mostly the phase of the cloud particles) and we see no difference in cloud optical depth between CMIP5 and CMIP6 then the only explanation is that there is no

difference in cloud microphysics/cloud-phase between the two ensembles. We do not see why that statement would contradict Hofer et al. (2019).

LN181: The twenty-year averaged cloud cover anomaly is compiled by a wide variety of circulation patterns. Only high frequency of a certain circulation pattern would depict the topography influence on the cloud cover composite. Thus, there is no information enough to infer the likelihood of circulation-driven cloud cover change.
We are not sure what the reviewer here means by saying that "only high-frequency …. of circulation pattern would depict the topography influence on cloud cover.".
We looked in detail at the cloud cover response for each of the 11 models chosen for downscaling. The overall message is that except for MIROC5, the individual models generally capture the ensemble mean really well. We have done this analysis for low-, mid- and high-level clouds, as well as for the total cloud cover. We have added the corresponding figures to the supplementary material of the manuscript (S13 and S14), as well as two sentences in the main text explaining why we think that the models capture the overall cloud cover response for CMIP5 and CMIP6.
Given that the individual models capture the overall ensemble mean cloud cover change well, we do not see why cannot make an inference about the topographic influence as described in Hahn et al. (2020) using similar MAR simulations. But maybe the reviewer did comment on a different aspect of our work here that we did not fully grasp.

Figure 5 and 6: consider two different color maps to stress the fact that colors shading in summer is not comparable with autumn. Perhaps, relative changes (e.g., ratio) instead of absolute changes could be here considered
The absolute magnitude can be seen from Fig.1 and Fig.2. In Figure 5 and 6 we are more interested in showing the pattern of changes and their physical explanation to what is causing them.

However, to avoid confusion for the reader we have added a sentence in Figure text 5 and 6 emphasizing the difference in colourbars of these two plots. Additionally, plots of the relative change for Figure 5 and 6 were added to the Supplementary Material (S.15 and S.16).

LN245: precipitation has so far been discarded of the analysis, but here it would be interesting to assess if precipitation, more specifically liquid precipitation, could play a role in the snow darkening and surface runoff.
Investigation of the rainfall has already been done before by Hofer et al. (2020), Fig. 6 (A-B). This figure shows that rainfall only really diverges towards the last two or three decades of the simulations between CMIP5 and CMIP6, but SMB starts diverging in the early 21st century (around 2020).

**Technical corrections**

LN9: spell the name of the regional climate model correctly
This was corrected in the revised manuscript.

LN11: indicate the corresponding level of uncertainty
We do not understand to what the referee thinks we should 'indicate the corresponding level of uncertainty' in LN11, as there is no result presented in LN11. The next closest value is presented in LN13 ("...during autumn with a reduction of 14.1 ± 4.8 mmW…") which already indicates an uncertainty.

LN27: spell the surname of the main author correctly
This was corrected in the revised manuscript.

LN32: spell the surname of the main author correctly
This was corrected in the revised manuscript.

LN51: downwards instead of "down towards"
This was changed accordingly in the revised manuscript.

LN51: LWU is defined as LWD
This was corrected in the revised manuscript.

LN55: introduce SWD at the beginning of the sentence
We have slightly adapted the wording of this sentence to avoid any confusion for the readers.

LN56: suggested place to define SMB instead of doing it in the Introduction
Equation 1 and the definition of SMB have been moved to the 'Methods' section in the revised manuscript.

LN59: spell the surname of the main author correctly
This was corrected in the revised manuscript.

LN71: spell the name of the regional climate model correctly
This was corrected in the revised manuscript.

LN154: Figure 2 c and d, instead of "a and b"
This was corrected in the revised manuscript.

LN156 Figure 2 c instead of "a"
This was corrected in the revised manuscript.

LN179: total instead of "toal"
This was corrected in the revised manuscript.

References

Bennartz, Ralf, et al. "July 2012 Greenland melt extent enhanced by low-level liquid clouds." Nature 496.7443 (2013): 83-86.

Box, Jason E., et al. "Greenland Ice Sheet Rainfall, Heat and Albedo Feedback Impacts From the Mid‐August 2021 Atmospheric River." Geophysical Research Letters 49.11 (2022): e2021GL097356.

Fausto, Robert S., et al. "The implication of nonradiative energy fluxes dominating

Greenland ice sheet exceptional ablation area surface melt in 2012." Geophysical Research Letters 43.6 (2016): 2649-2658.

Hofer, Stefan, et al. "Cloud microphysics and circulation anomalies control differences in future Greenland melt." Nature Climate Change 9.7 (2019): 523-528.

McCrystall, Michelle R., et al. "New climate models reveal faster and larger increases in Arctic precipitation than previously projected." Nature communications 12.1 (2021): 6765.

Tedesco, Marco, et al. "The role of albedo and accumulation in the 2010 melting record in Greenland." Environmental Research Letters 6.1 (2011): 014005.

Tedesco, Marco, et al. "Arctic cut-off high drives the poleward shift of a new Greenland melting record." Nature Communications 7.1 (2016): 11723.

Wang, Wenshan, et al. "Greenland surface melt dominated by solar and sensible heating." Geophysical Research Letters 48.7 (2021): e2020GL090653.

References

Bennartz, Shupe, M. D., Turner, D. D., Walden, V. P., Steffen, K., Cox, C. J., Kulie, M. S., Miller, N. B., & Pettersen, C. (2013). July 2012 Greenland melt extent enhanced by low-level liquid clouds. Nature (London), 496(7443), 83–86. https://doi.org/10.1038/nature12002

Box, Wehrlé, A., As, D., Fausto, R. S., Kjeldsen, K. K., Dachauer, A., Ahlstrøm, A. P., & Picard, G. (2022). Greenland Ice Sheet Rainfall, Heat and Albedo Feedback Impacts From the Mid‑August 2021 Atmospheric River. Geophysical Research Letters, 49(11), n/a–n/a. https://doi.org/10.1029/2021GL097356

Fausto, As, D., Box, J. E., Colgan, W., Langen, P. L., & Mottram, R. H. (2016). The implication of nonradiative energy fluxes dominating Greenland ice sheet exceptional ablation area surface melt in 2012. Geophysical Research Letters, 43(6), 2649–2658. https://doi.org/10.1002/2016GL067720

Hansen, Simonsen, S. B., Boberg, F., Kittel, C., Orr, A., Souverijns, N., Van Wessem, J. M., & Mottram, R. (2022). Brief communication: Impact of common ice mask in surface mass balance estimates over the Antarctic ice sheet. The Cryosphere, 16(2), 711–718. https://doi.org/10.5194/tc-16-711-2022

Hofer, Lang, C., Amory, C., Kittel, C., Delhasse, A., Tedstone, A., & Fettweis, X. (2020). Greater Greenland Ice Sheet contribution to global sea level rise in CMIP6. Nature Communications, 11(1), 6289–11. https://doi.org/10.1038/s41467-020-20011-8

Kjeldsen, Khan, S. A., Colgan, W. T., MacGregor, J., & Fausto, R. S. (2020). Time-Varying Ice Sheet Mask: Implications on Ice-Sheet Mass Balance and Crustal Uplift. Journal of Geophysical Research. Earth Surface, 125(12), n/a–n/a. https://doi.org/10.1029/2020JF005775

McCrystall, Stroeve, J., Serreze, M., Forbes, B. C., & Screen, J. A. (2021). New climate models reveal faster and larger increases in Arctic precipitation than previously projected. Nature Communications, 12(1), 6765–12. https://doi.org/10.1038/s41467-021-27031-y

Mostue. (2022). Greenland Surface Energy Budget Response in CMIP6. https://www.duo.uio.no/handle/10852/93884

Tedesco, Fettweis, X., van den Broeke, M. R., van de Wal, R. S. W., Smeets, C. J. P. P., van de Berg, W. J., Serreze, M. C., & Box, J. E. (2011). The role of albedo and accumulation in the 2010 melting record in Greenland. Environmental Research Letters, 6(1), 014005. https://doi.org/10.1088/1748-9326/6/1/014005

Tedesco, Mote, T., Fettweis, X., Hanna, E., Jeyaratnam, J., Booth, J. F., Datta, R., & Briggs, K. (2016). Arctic cut-off high drives the poleward shift of a new Greenland melting record. Nature Communications, 7(1), 11723–11723. https://doi.org/10.1038/ncomms11723

---

## Author Comment (AC2)

**Answer to Referee #2 – Manuscript tc-2023-24**

Review Cloud- and ice-albedo feedbacks drive greater Greenland ice sheet sensitivity to warming in CMIP6 than in CMIP5.

I've read the manuscript with interest, and the study is publishable after some questions are properly addressed. It analyses in detail the differences between projections for the Greenland Ice Sheet from CMIP5 and CMIP6 models. This has not been done before, and the study provides interesting new findings. However, addressing the issues below can strengthen the study and can take away potential concerns of readers.

We would like to thank Referee #2 for taking the time to read our manuscript so thoroughly and provide in-depth feedback on our study. Below we have responded to their comments and how we would like to address them. We think that our manuscript will greatly benefit from the adjustments.

Main issues:

1: The focus on the radiative terms only.

With some references, the authors justify why only the radiative terms of the surface energy balance (SEB) are discussed in this manuscript. I'm not a prior convinced that this is justified. The sensible heat flux (SHF) is a significant contributor to melt in the ablation zone (you can find many papers about that), and the ablation zone is where the differences are made. I, therefore, ask the following:

1. a) The authors repeat the analysis as shown in figure 2a and 2b and figure 5 for SHF, LHF (latent heat flux) and GHF (ground heat flux - the residual most likely). If these terms are indeed insignificant, as the authors argue now, these figures may be added to the SOM and single references like "we've studied these terms too and they are insignificant" will do in the main text. However, if SHF/LHF/GHF changes are not negligible, their discussion needs to be included in the text.

   We have studied the SHF and LHF and they are insignificant compared to the other energy fluxes in MAR, especially when considering strong warming scenarios like SSP5-8.5 and RCP8.5. We have commented on this in the revised manuscript and added the plots for these terms in the Supplementary Material S.4.

   During melt, MAR simulates GHF close to zero.Therefore, we did not investigate this term further. We have included this information in the revised manuscript for clarity.

It might also add more clarity to the cloud arguments, as SHF, in contrast to LWD, is not influenced by changes in cloud cover. Conversely SHF is also influenced by surface warming too, I'm not sure a prior if changes in SHF are clearer than the LWD change. Well, the authors have to find out.

Given that the sensible heat flux is proportional to the gradient between surface temperature and near-surface air temperature one would assume that the SHF is also influenced by

changes in cloud cover, as shortwave radiation tends to warm darker surfaces more than brighter surface, in contrast to longwave radiation. Clouds do indeed shift the surface energy budget terms from more shortwave towards more longwave radiation, changing the surface temperature distribution. Furthermore, please see our previous comment why we did not include a more in-depth description of the sensible heat flux in our manuscript.

1. b) Consequently Equation 2, the SEB, should be adjusted to

ME = LWD - LWU + SWD(1 - \alpha) + SHF + LHF + GHF [W /m2]

Please remove the \epsilon \sigma T^4 term from the equation, as this equation is never used in the manuscript (and please remove the crosses as that denotes the outer product of matrices), this relation can be mentioned in the running text; and please correct the units. Finally, the equation as a whole is the SEB, not the right-hand side.
We agree with the referee and have changed Equation 2 accordingly. The units have also been corrected in the revised manuscript.

2: The role of melt water buffering by firn:

In brief, the manuscript now states this: The CIMP6-CMIP5 change in SMB for a given warming (fig. 1a) is occurring in fall (fig. 1c), while the fall SEB is virtually unchanged (fig 2b, 5-right row). However, the CMIP6 fall albedo of the ablation zone is lower, leading to more melt which, in absence of firn, runs off directly (fig 6 - right column). Correct, however, by zooming in on the fall, it ignores the relevance of the melt-refreezing-runoff pattern in summer. Furthermore, I'm highly puzzled by the lack of refreezing increase during any phase of the warming. This runoff increase is the overarching feature of all RACMO simulations and very visible (as far as I know) in MAR simulations driven by reanalyzes. Therefore, I ask the following.

1. a) A description of the exact run protocol of MAR needs to be added in the methods section. Furthermore, (as understanding the firn response is important in this study), expand the description of the firn model with details relevant for this study. For example, how thick can the firn column become in MAR? Hence, the authors need to be able to address to which extent coding choices have impacted the modelled melt water buffering capacity in a transient climate?
Only the 30 first meter of snow is resolved in these simulations. A layer is automatically added/removed at the bottom if the total snow is < 29m or > 31m The maximum liquid water content in MAR is 7% vs 5% in HIRHAM vs 2% in RACMO. The choice of 7 % in MAR is discussed in Lefebre et al. (2003). Therefore, the densification and the warming of the snowpack is faster in MAR than in the other models. It also means the meltwater capacity retention of the snowpack decrease faster in MAR. Finally, the same MAR executable has been used in both CMIP5 and CMIP6 simulations.

*We have extended the 'Methods' to include the run protocol for MAR in the revised manuscript.*

The authors now state in lines 248-254 that the faster warming in CMIP6 induces that the percolation has more remaining melt water buffering capacity when the simulation crosses the 4 K warming point. I'm not sure if that is correct, as many MAR projections available nowadays are not run in single linear mode, thus one single MAR realization that started in 1950 and ended in 2100. Those many MAR projections that are in the community are run with a "each year initialized separate"-protocol; thus, that the weather of that year is repeated until the MAR firn column, and hence the SMB and its terms, have become in equilibrium.

1. b) If the latter protocol has been used, the 'faster warming' argument is invalid. Still, a detailed explanation of figure 6, left column, needs to be given. If the first protocol has been used, it is worth to show the difference in firn air content (or another firn state metric) to highlight that this different firn state strongly contributes to the pattern visible in the left column of figure 6. Furthermore, and that is really important IMHO, the conclusions should then be: CMIP6 melt more due to stronger warming and, our new point, decreasing (high) cloud cover, both only partly mitigated by more remaining melt water buffering capacity due to the faster pace of warming.
   *MAR has been run in "community" mode meaning that a member is started every 5 years over 1950-2090 and initialised by the snowpack simulated for this year by former MARv3.9 based simulations using the same GCM as forcing. Each member simulates at least 15 years (including 10 years of spin-up). As the period simulated by each member covers at least two members initialised at different years (5 and 10 years ago), the retained years of each member has been chosen to be independent of the initial conditions i.e. to have difference of SMB, runoff and refreezing lower than 1 GT/yr between the different members for the same year. Due to the high liquid water content allowed in MAR, a snowpack can lose quickly (~10years) its capacity to retain meltwater as it becomes too dense.*

2. d) The authors should discuss in more detail why not part of the melt increase is buffered by runoff (figure 1), contrary to findings in preceding studies (like Noël 2021, doi: 10.1029/2020GL090471). From Figure S7, bottom row, I would expect a clearer visible increase in refreezing. Or is the increase in refreezing matched by the increase in rain? If so, please consider adding rain and runoff in Figure 1. Furthermore, is runoff indeed zero in the interior of Greenland? From Figure S7 it is not 100% clear.
   *We are not sure what the reviewer means by "melt buffered by runoff" in this case?*

   *Regarding adding rain and runoff, this had already been done in Hofer et al. (2020) Fig. 6 (A-B). This figure shows that rainfall only really diverges towards the last two or three decades of the simulations between CMIP5 and CMIP6. Additionally, the reason why refreezing in Fig. S7 bottom row is not visible compared to the other surface mass balance components is because compared to the other terms refreezing in summer (JJA) is almost an order magnitude smaller (at least in MAR)*

and melt really dominates all other terms (as it has also done for recent extreme melt years already).

3. d) The discussion of the very different impact of clouds changes over the ablation zone, compared to the percolation/accumulation zone, could be much stronger and clearer if in the SEB analysis (figure 2) the ablation zone and accumulation zone are separated. Most likely a static separation mask is much easier than a transient mask. A good mask is to take the ablation zone outline for 4 K warming, as that is the "warming" point in time that is analyzed most in the paper.
We agree with the reviewer that this would be an interesting addition to the study. In the revised manuscript, we will perform an additional SEB analysis for a division between the ablation zone and accumulation zone for a 4 degrees Celsius warming.

3: Don't leave uncertain things you could verify in the model data:

On numerous points (about 10-20, I lost count) the authors are unsure as they use "we argue" (line 181), "we expect" (line 124), "can possibly be explained" (line 248), "we suspect" (line 267) or "we suggest" (line 270), "likely due" (line 277) while the answer can be found in the model data the authors should have. Go and check your ideas in the model data and write with certainty when it is true and remove the statement if it is untrue. I don't see a valid reason to be unsure. When some of those "unsure statements" cannot be verified and are retained in the manuscript, please address the reasons in the reply to this review.
We thank the referee for this comment. We have made changes to the revised manuscript to avoid leaving the readers with any uncertainty regarding our statements. However, we would also like to refute the claim that everything that can be checked in model data has to be communicated with absolute certainty. First, authors have different styles of writing and we strongly believe that these phrasings fall within that category. Second, over the last decades many findings in climate science have naturally been refined, sometimes to a point where old knowledge has been fully replaced with new findings. Therefore, we think using slight nuancing in scientific communication can also future-proof the communicated findings. However, it all boils down to personal preference.

4: Demonstrate that these results are not coincidental by the CMIP5 & CMIP6 model selection but a genuine difference between CMIP5 and CMIP6:

1. a) Hofer et al 2020 gives the arguments for the model selection. This information needs to be summarized in this paper as it should not be necessary to read Hofer et al 2020 to understand this model selection.
We have included a summary of the reasoning behind the model selection by Hofer et al 2020 in the revised manuscript.

2. b) In the discussion, it needs to be showed (as good as possible) that these 5 & 6 models are representative for change in the modelled cloud climatology over Greenland. It (representativeness for cloud cover changes) is not mentioned in Hofer et al, 2020. I know this can be a lot of work (as modelled cloud cover over Greenland

from ~30 models needs to be compared), so I can understand if the authors use existing studies to demonstrate this - if these are available. Nonetheless, the authors make implicitly this generalization, however, it should be justified.

We agree with the reviewer that we implicitly assume that our subset of models is representative of the overall climate of the relevant CMIP ensemble. While the models have been selected with great rigor for the CMIP5 models (see Barthel et al. (2020)), the same cannot be said about the subsample of CMIP6 models. However, in Hofer et al. (2020) it has been shown that the CMIP6 models represent the general climate and warming trends for the Arctic quite accurately. In addition, we looked in detail at the cloud cover response for each of the 11 models chosen for downscaling. The overall message is that except for MIROC5, the individual models generally capture the ensemble mean really well. We have done this analysis for low-, mid- and high-level clouds, as well as for the total cloud cover. We have added the corresponding figures to the supplementary material of the manuscript (S13 and S14), as well as two sentences in the main text explaining why we think that the models capture the overall cloud cover response for CMIP5 and CMIP6.

Minor general point on units:

1. a) I would prefer that K (Kelvin) is used instead of degrees Celsius.

   As either of these units are equally accessible to the audience and using either does not change the interpretation of our results we chose to not change our results to K (Kelvin).

2. b) Albedo is unitless, and percentual changes of albedo are meaningless. Please correct this in Figures 7, S9 and S10, and section 3.4. Especially the numbers in lines 274-275 are wrong wrt units.

   Albedo is a fractional quantity that by definition can take up values between 0 and 1 (fraction) - but in mathematics percentage is not seen as a unit, but rather as a different way of expressing a fraction (½, 0.5 or 50% all mean exactly the same thing). However, to avoid any confusion we have changed the indicated text and figures.

3. c) Use hPa and not Pa in lines 184-185.

   This was changed accordingly in the revised manuscript.

4. d) Equation 1 has units Gt yr, but no result is shown with that unit. mmWE yr-1 or mmWE season-1 is used everywhere. So why not for Equation 1 too? At the other hand, Gt yr-1 is a unit easier to interpret for a larger audience, so it is worth to consider to use this unit more often in the manuscript - like in the running text.

   We agree with the referee that the units should be consistent throughout the manuscript. We have changed the unit from 'mmWE' to 'Gt yr-1' for Figure 1, 2a, 2b, and for the corresponding results in the running text of the revised manuscript.

Minor comments:

13: Please remove these two numbers (4.8 & 12.5 mmWE) as they make no sense for anyone without more context. Alternatively, use Gt yr-1 here. But in case of the latter, Gt yr-1 should be used more often in the manuscript - as already stated above.
We have changed the units from mmWE to Gt yr-1 accordingly in the revised manuscript.

19: "Undocumented" Please rephrase by e.g. "unnotified". Those changes have been documented before - as they were in the output files of many simulations - but nobody has written a paper about it before.
This was changed accordingly in the revised manuscript.

24: Add Noël and van Kampenhout, 2021 (cited above) here.
Noël and van Kampenhout 2021 was added accordingly in the revised manuscript.

32: "Broek" is "Broeke". Check also other the references on typos.
Thank you for pointing this out, this was corrected in the revised manuscript.

51: The second LWD must be LWU.
Thank you, you are correct. This was corrected in the revised manuscript.

77: "In turn" is IMHO not the right connection word here.
We have removed 'In turn' from the revised manuscript.

83: Add a bit more detail - is this a 4 K a (near) surface (T0m, T2m) of, e.g., 500 hPa, warming? Is the year chosen for the GrIS as a whole or for each grid point separately? And what does it imply for the "consistency of the results" as the JJA and SON 4 K warming frames does not refer to the same years and thus firn state?
We use a 4 degree Celsius near surface warming (T2m) for the GrIS as a whole. We have added more detail to that part of the revised manuscript.

Since the IPCC AR6, some of the critiques that have come out state that you cannot directly compare CMIP5 and CMIP6 models because they have different warming rates and some might be too sensitive to greenhouse gas forcing. Therefore, we have focused on looking at differences for a given warming rate. However, we agree with the comment and would welcome any suggestion on how to treat the different firn states for the different seasons.

103: The exact locations of the model edges, expressed in lat & lon, give a reader little clue. Add a figure of the domain extend (in the SOM), or write it down in words, e.g. "extends till Svalbard", "extends xxx km" to all sides of Greenland", …
We have added a figure of the research domain to the revised Supplementary Material (S.1).

106: Please add the typical elevations of the lowermost model layers.
The 5 lower atmospheric vertical levels are ~20, 12.0, 8.0, 4.0, 2.0 m above surface. This was added to the description of MAR in the 'Methods' of the revised manuscript.

113: "near surface temperature anomaly [C (or K)]"
This was corrected accordingly in the revised manuscript

118: Consider not to start a new paragraph here.
This was changed accordingly in the revised manuscript.

Figures 1, 2, 3: Labels have turned into white in the official pdf. Ensure this is ok in the revised manuscript. Furthermore, if you consider a SMB anomaly for a season, the unit is mmWE season-1, not mmWE. I'm not aware of a nice abbreviation for season (like yr and s), so possibly the authors may conceive one.'
Thank you for pointing this out. We will ensure that all labels are visible in the revised manuscript. We have changed the unit for the SMB anomaly accordingly from 'mmWE ' to 'mmWE season-1'.

171: Please state more clearly at the end of the paragraph that despite the changes in Figs 2d, 3b, 4-"right", there is no discernable change in Fig 2b that explains Figure 1c.
We have changed the wording slightly to make this more clear for the readers.

241: "we do not see the same buffering effect". Please mention that this is due to the absence of significant melt in fall in the percolation zone. There is no water to buffer.
We agree and have changed the wording from "In autumn (SON) we do not see the same buffering effect of more refreezing in MAR CMIP6 in the percolation zone as we saw for summer (JJA)." to "In autumn (SON) we do not see the same buffering effect of more refreezing in MAR CMIP6 in the percolation zone as we saw for summer (JJA), partly due to a decrease in meltwater production in this region."

Figure 8: I like the idea of an explaining figure. This one, however, is very unclear, thus fails to meet its aim. It should show that that in summer, the LW effect dominates over the SW effect in the percolation and accumulation zone due to the high albedo of snow, while SW dominates in the ablation zone due to the lower albedo. Improve the figure or remove it.
We have made an improved version of Figure 8 in the revised manuscript, highlighting that SW radiation dominates over the darker surfaces and LW radiation over brighter surfaces in summer.

Citation: https://doi.org/10.5194/tc-2023-24-RC2

References

Barthel, A., Agosta, C., Little, C. M., Hattermann, T., Jourdain, N. C., Goelzer, H., Nowicki, S., Seroussi, H., Straneo, F., and Bracegirdle, T. J.: CMIP5 model selection for ISMIP6 ice sheet model forcing: Greenland and Antarctica, The Cryosphere, 14, 855–879, https://doi.org/10.5194/tc-14-855-2020, 2020.

Hofer, Lang, C., Amory, C., Kittel, C., Delhasse, A., Tedstone, A., & Fettweis, X. (2020). Greater Greenland Ice Sheet contribution to global sea level rise in CMIP6. *Nature Communications*, *11*(1), 6289–11. https://doi.org/10.1038/s41467-020-20011-8

Lefebre, F., H. Galle´e, J.-P. van Ypersele, and W. Greuell, Modeling of snow and ice melt at ETH Camp (West Greenland): A study of surface albedo, J. Geophys. Res., 108(D8), 4231, doi:10.1029/2001JD001160, 2003.

Noël, van Kampenhout, L., Lenaerts, J. T. M., van de Berg, W. J., & van den Broeke, M. R. (2021). A 21st Century Warming Threshold for Sustained Greenland Ice Sheet Mass Loss. *Geophysical Research Letters*, *48*(5), n/a–n/a. https://doi.org/10.1029/2020GL090471